# Multifunction waveform generator for EM receiver testing

Kai Chen, Sheng Jin, Ming Deng

China University of Geosciences, Beijing, China

*Correspondence to:* Kai Chen (ck@cugb.edu.cn)

**Abstract.** In many electromagnetic (EM) methods, such as magnetotelluric, spectrum induced polarization, time domain induced polarization, and controlled source audio magnetotelluric methods, it is important to evaluate and test the EM receivers during their development stage. To assess the performance of the developed EM receivers, controlled synthetic data that simulates the observed signals in different modes is required. Based on our testing, the frequency range, frequency precision, and time synchronization of the currently available function waveform generators in the market are deficient. This paper presents a multifunction waveform generator with three waveforms: 1) a wide-band low-noise electromagnetic field signal to be used for magnetotelluric, audio-magnetotelluric, and long period magnetotelluric studies; 2) a repeating frequency sweep square waveform for controlled source audio magnetotelluric and spectrum induced polarization studies; and 3) a "positive-zero-negative-zero" signal that contains primary and secondary fields for time domain induced polarization studies. In this paper, we provide the principles of the above three waveforms along with a hardware design for the generator. Furthermore, testing of the EM receiver was conducted with the waveform generator, and the results of the experiment were compared with those calculated from the simulation and theory in the frequency band of interest.

Keywords: Multifunction waveform generator; EM receiver; pseudo-random binary sequence; chopper; signal synthesiser

## 1 Introduction

Electromagnetic (EM) methods are successfully used in a variety of applications, including metal ore investigations, ground water explorations, hydrocarbon prospecting, volcano research, and deep earth research (Wei et al., 2010; Key, 2003). In these applications, EM methods are used to measure natural

or controlled source signals, and then to interpret the underground electrical structure using data processing and inversion techniques (Osinowo and Olayinka, 2012; Scheuermann, 2016). There are many EM methods in use today; however, in this paper we are interested in magnetotelluric (MT) (Cagniard, 1953), controlled source audio magnetotelluric (CSAMT) (Sandberg and Hohmann, 1982), spectrum induced polarization (SIP) (Johnson, 1984), and time domain induced polarization (TDIP) (Marshall and Madden, 1959) methods. The requirements for field instruments supporting these methods include high resolution, large exploration depth, low cost, and high efficiency field data acquisition. All of the above EM methods are dependent on the quality of the field data acquired by the EM instrument. Therefore, the specifications of the EM receiver are of particular importance for high quality EM prospecting. The current EM receivers, such as the V8 receiver from Phoenix Geophysics (PhoenixGeophysics, 2017), GDP32 from Zonge (Zonge, 2017), ADU-07e from Metronix (Metronix, 2017), and KMS-820 from KMS (KMS, 2017), are all specified as being multifunction, multi-channel, and easy to use with low noise levels and low clock drift errors.

The China University of Geoscience began developing a multifunction EM receiver (EMR6) in 2014, which is mainly intended for EM field measurements in surface and tunnel. The receiver support method modules include the MT, CSAMT, SIP, and TDIP methods. In addition, the MT function includes the audio magnetotelluric (AMT), magnetotelluric, and long period magnetotelluric (LMT) methods in different frequency ranges. During the development of the instrument, the receiver was tested with each method module after the electrical parameters were characterised. For the MT method module test (Ge et al., 2016), a pseudorandom bit sequence (PBRS) module was designed, which included a white noise source that simulated a broadband natural source MT signal in the 5 x $10^{-4}$ Hz to 14 kHz frequency range, which was suitable for the AMT and MT mode. However, the lower frequency range for the LMT test was deficient. For the CSAMT and SIP mode tests, which require a repeating frequency sweep square waveform, the test signal should be time synchronized and repeated. There are function waveform generators available in the market, such as the Agilent 33250A, which support multiple broadband waveforms; however, the frequency precision and time synchronization error in these generators are deficient. For example, in the TDIP mode test, the arbitrary waveform function is suitable, but the time synchronization error is deficient.

We developed a special multifunction waveform generator to meet the testing requirements of the EMR6 receiver. The waveform generator supports a broadband low-noise pseudo-random binary

sequence (PRBS) for MT, repeating frequency sweep square waveform for CSAMT and SIP with a programmable frequency step list, and "positive-zero-negative-zero" (PNZN) waveform containing both primary and secondary fields for the TDIP mode. Moreover, additional requirements for the waveform generator included: 1) ease of use, auto repeat circulate output according the scheduled frequency step list; 2) high precision phase and time synchronisation; 3) low power consumption for the LMT mode test, which has the capacity of the built in Li-ion battery that allows for one week of operation.

This paper presents the details of the generator, including both the system design and the hardware principles. The results of the experimental testing using the EMR6 are also provided.

## 2 System design

The multifunction waveform generator is designed to output three types of waveforms: white noise for MT (AMT/MT/LMT), a repeated swept square waveform for CSAMT and SIP modes, and a "PNZN" waveform for the TDIP mode.

The MT method consists of three branches – AMT, standardised MT, and LMT – and these branches differ in terms of the exploration depths of interest and the effective frequency ranges. Figure 1 illustrates the frequency ranges applicable to the three branches of the MT method. The high frequency band in the AMT method is from 10 kHz to 0.1 Hz, which spans five decades; the frequency band in the standardised MT method is from 320 Hz to 5 x $10^{-4}$ Hz, which spans approximately six decades; the frequency band in the LMT method is from 1 Hz to 1 x $10^{-5}$ Hz, which spans approximately five decades. A PRBS generator is the best choice for generating broadband signals (Amrani et al., 1998). The highest frequency is determined by the width of the smallest encoding, and the lowest frequency depends on the length of the PRBS. According to the three different frequency band modes, the length of the PRBS must be greater than 1 x $10^6$. Therefore, we designed a PRBS with length $2^N - 1$, where N equals 24 and the length is 16 M. By changing the smallest code width of the PRBS, it was easy to meet the three MT modes operating in different frequency ranges.

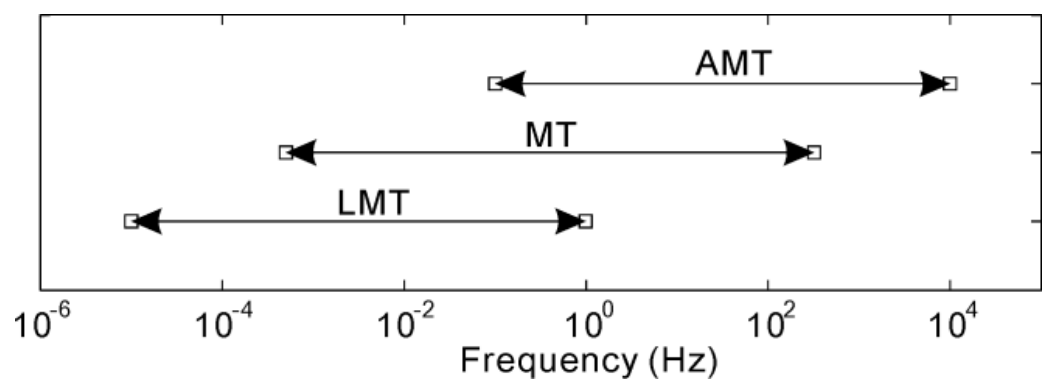

**Figure 1: Three MT sounding methods with different frequency ranges.**

The formal definition of the PRBS is:

$$D = X^{24} + X^{7} + X^{2} + X + 1. \quad (1)$$

The length of the sequence is 16 M. The smallest code width, which changes for different fundamental frequencies, can be selected from 10 μs, 1 ms, and 100 ms. Table 1 lists the parameters for the three modes.

**Table 1. PRBS Parameters for three MT modes.**

| Mode | Length | Smallest code width | Available frequency range |
|---|---|---|---|
| AMT | 16 M | 10 μs | 100 kHz to 6 mHz |
| MT | 16 M | 1 ms | 1 kHz to 60 μHz |
| LMT | 16 M | 100 ms | 10 Hz to 0.6 μHz |

The designed frequency range for the CSAMT method is from approximately 0.1 Hz to 10 kHz. The programmable frequency stepping schedule is according to the typical CSAMT frequency stepping schedule used in the field when working with V8 receiver from Phoenix Geophysics. The frequency stepping schedule contains 41 frequency points, and lasts for 50 min. Figure 2 presents the typical frequency stepping schedule plotted on a log scale, and two frequency points in a double frequency range as an approximate average in the log scale. The 41 frequency points cover four decades and extend from 9,600 Hz to 0.9375 Hz. All frequency points are derived from a 12.288 MHz high-stability clock source. To increase the lower frequency signal-to-noise ratio (SNR), the length of the stacking time of the lower frequencies is longer than that of the higher frequencies. The longest stacking time is 323 s for 1.25 Hz in the low frequency band, and the shortest is 40 s in the high frequency band.

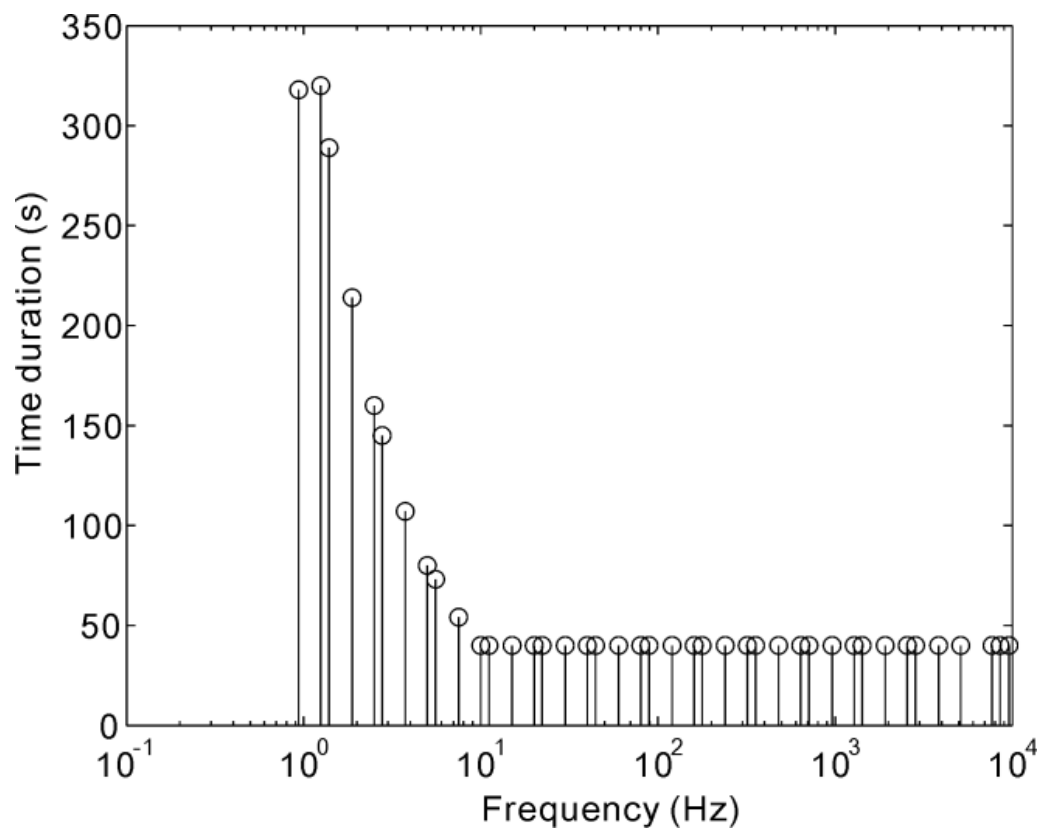

**Figure 2: Stacking time length for a typical CSAMT frequency stepping schedule.**

The designed frequency range for the SIP method is from 128 Hz to 0.0625 Hz. The frequency stepping schedule is the same as the typical SIP step schedule used in the field when working with V8 receiver from Phoenix Geophysics. Figure 3 shows a typical frequency stepping schedule plotted on a log scale, and frequency stepping by double. There are 12 frequency points, each of which last for 15 min, and the 12 points cover approximately four decades and extend from 128 Hz to 0.0625 Hz. As was the case in the CSAMT mode, all frequency points could be derived from a 12.288 MHz clock source. To increase the lower frequency SNR, a longer stack time was used in the lower frequency band versus that used in the higher frequency band. The longest stack time was 273 s in the low frequency band at 1.25 Hz, and each point lasted for 50 s in the high frequency band.

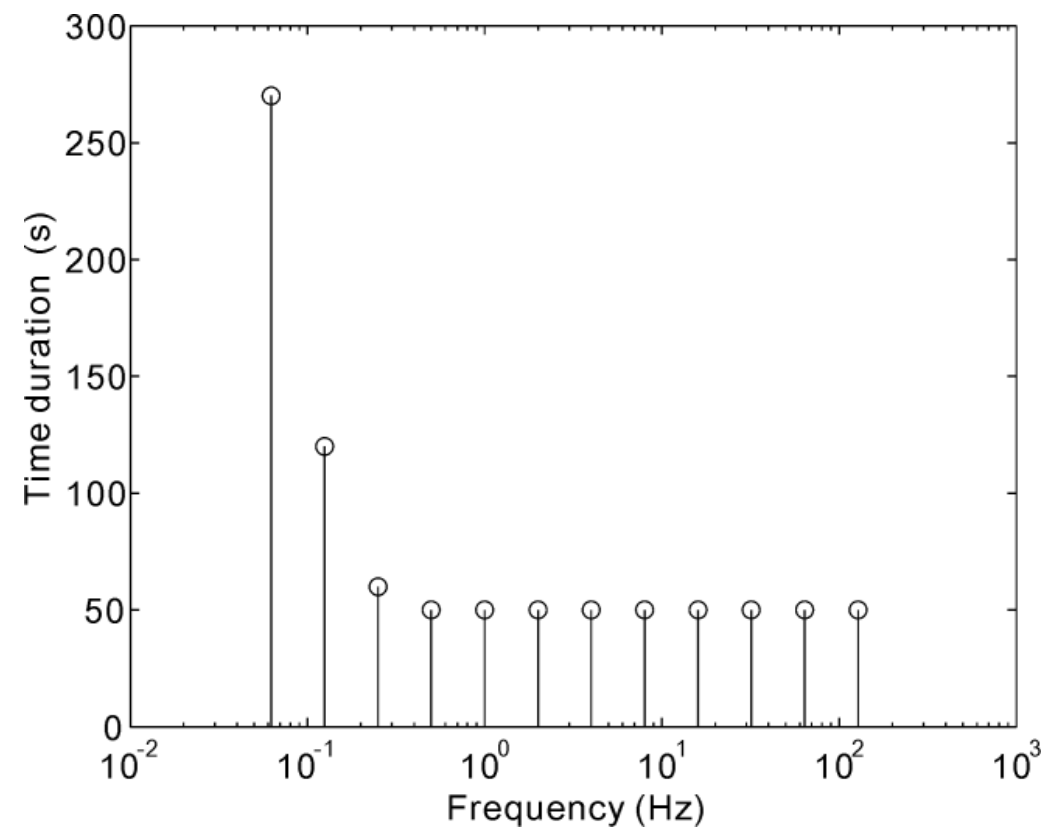

**Figure 3: Stack time length for a typical SIP frequency stepping schedule.**

The PZNZ waveform for the TDIP module is shown in Fig. 4 and consists of four phases: positive ON time, OFF time, negative ON time, and OFF time. The duty ratio is 1:1. In the figure, T denotes the four-phase period and the width of the pulse is T/4. The term A denotes the amplitude of the primary electrical field, B denotes the maximum amplitude of the secondary electrical field, and A + B denotes the total electrical field. The self-potential and other disturbances are not considered in the figure.

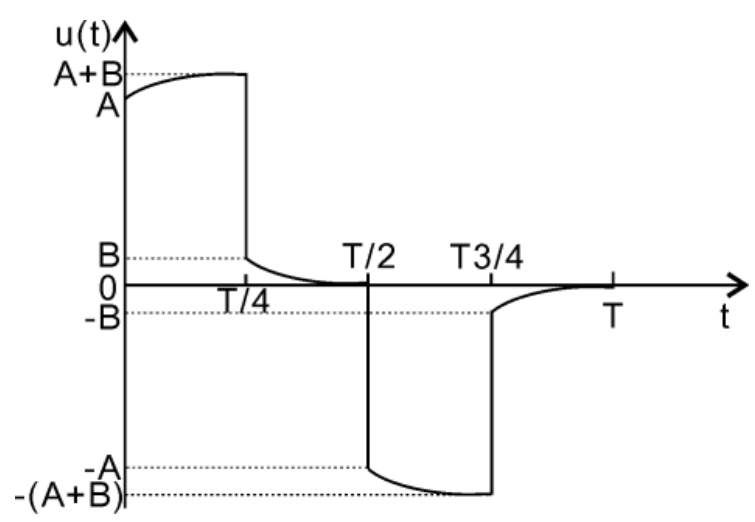

**Figure 4: PZNZ waveform from a theoretical simulation.**

$$u_1(t) = \begin{cases} A,\ 0 \le t < T/4 \\ 0,\ T/4 \le t < T/2 \\ -A,\ T/2 \le t < 3T/4 \\ 0,\ 3T/4 \le t < T \end{cases} \quad (2)$$

$$u_2(t) = \begin{cases} B - Be^{-t/\tau},\ 0 \le t < T/4 \\ Be^{-(t-\frac{T}{4})/\tau},\ T/4 \le t < T/2 \\ Be^{-(t-\frac{T}{2})/\tau} - B,\ T/2 \le t < 3T/4 \\ -Be^{-(t-\frac{3T}{4})/\tau},\ 3T/4 \le t < T \end{cases}, \quad (3)$$

where $u_1(t)$ and $u_2(t)$ denote the primary and secondary electrical fields, respectively, and the exponent attenuation curve refers to the secondary electrical field.

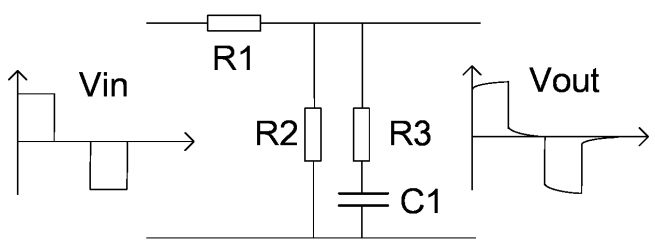

**Figure 5: Schematic of the PZNZ waveform generator.**

Figure 5 shows a schematic of the circuit used to generate the PZNZ waveform, which contains both primary and secondary electrical fields. The resistor-capacitor (RC) network transforms the PZNZ ($V_{in}$) waveform into a PZNZ waveform ($V_{out}$) with the secondary electrical field.

$$\mathrm{A} = \frac{R_3//R_2}{R_1+R_3//R_2}K \quad (4)$$

$$\mathrm{B} = \frac{R_3//R_2}{R_1+R_3//R_2}K - \frac{R_2}{R_1+R_2}K \quad (5)$$

$$\tau = \mathrm{R}_3 C_1 \quad (6)$$

The amplitude of Vin is ±K. A and B used in Eqs. (2) and (3) can be calculated from Eqs. (4) and (5). The time constant (τ) is given by Eq. (6).

## 3 Hardware principle

### 3.1 Block diagram

Figure 6 presents a block diagram of the waveform generator, which has the advantages of automation, ease of use, high phase precision, and low power. As shown in the figure, the hardware consists of an input switch, microcontroller unit (MCU), GPS module, complex programmable logic device (CPLD), compensated microprocessor crystal (MCXO), real time clock (RTC), chopper, shaper, multiplexer, power conversion circuit, and built-in Li-ion battery package. The input switch is used to change the work mode without configuring the external complicated parameters. The LED is used to indicate the working status. The different work modes output different LED flash patterns. The GPS module is LEA-6T from U-blox, which provides a high precision time pulse per second with low power

consumption. The MCXO has the specifications of high stability clock source (12.288 MHz) (±30 ppb) and low power consumption (3.3 V &12 mA). To lock the GPS, the MCU receives the time information from the GPS module and writes to the RTC. The CPLD is used to implement a frequency divider, logic operator, PPS lock, and tracking. The RTC is the time counter used for circulation of the frequency stepping schedule. The chopper circuit chops a high precision DC reference into a bipolar square waveform under the control of the CPLD. The shaper generates the TDIP PZNZ waveform. Moreover, the sum of the primary and secondary field signals is also provided as output. The multiplexer is controlled by the MCU to select either the chopper or the shaper output. The power module converts the Li-ion battery (11.1 V and 10 Ahr) voltage to digital power at 3.3 V and analogue power at ±3 V.

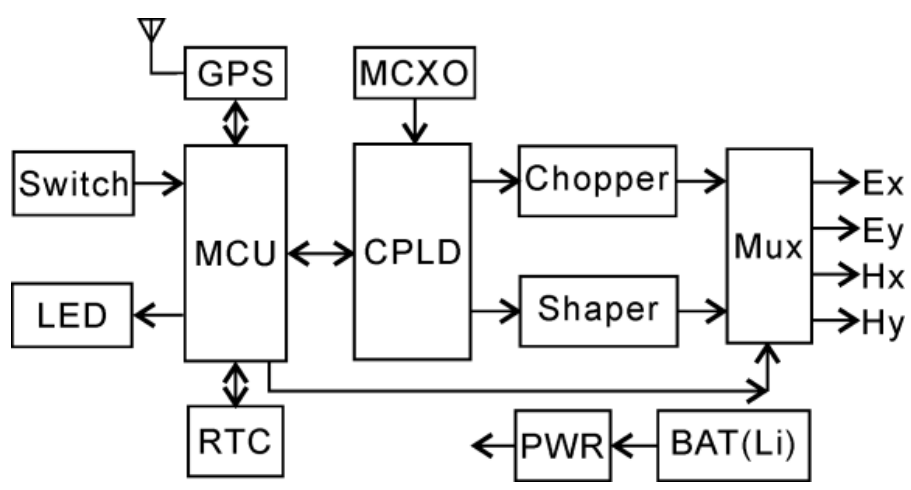

**Figure 6: Block diagram of the multifunction waveform generator circuit.**

When the MT mode is selected, the MCU controls the CPLD to generate two independent PRBS. The outputs Ex and Hy share one PRBS, whereas Ey and Hx share another PRBS. The two PRBSs are different for different phases. The chopper circuit converts the 3.3 V COMS PRBS into a bipolar ±10 mV square wave for Ex and Ey. The amplitude of Hx and Hy is ±100 mV. The MCU uses the multiplexer output to select the chopper output. Based on the different modes (AMY/MT/LMT), the output selected by the input switch and the smallest code width of the PRBS is changed. To decrease the power consumption, the GPS module is powered down during the MT mode.

When the CSAMT mode is selected, the MCU reads the GPS time information and writes to the RTC. The MCU controls the CPLD divisionde factor according to the frequency stepping schedule. The CPLD divides the clock source (12.288 MHz) to the target frequency step by step. The divider is trigged by the PPS from the GPS module. The 41 frequency point clock signal drives the chopper circuit to generate a bipolar square waveform. All frequency stepping schedules start from the reference time base of 00:00:00 and repeat from high frequency to low frequency. For example, after the power is

turned on and the GPS is locked, the current time is 02:20:00, the period of frequency stepping schedule is 50 min, the residual time is 40 min, the first output signal is the No. 40 frequency point (1.25 Hz), and the output is the rest 10 min of the current schedule. The frequency stepping schedule is repeated continually.

The procedure in the SIP mode is the same as that described above for the CSAMT module for the step schedule with time synchronization. A typical frequency stepping schedule period is 15 min, and contains 12 frequency points.

When the TDIP mode is selected, the RTC time is locked to the GPS module and the CPLD generates pulse with a 2 s width synchronization as the PPS from the GPS module to drive the shaper circuit. The reference time base is 00:00:00, and the typical period is 8 s with duty ratio 1:1.

### 3.2 Clock source

In accordance with the high precision phase requirement from the CSAMT and SIP modes, an integrated U-blox GPS module and MCXO were selected as the high stability clock source. The LEA-6T module series is a family of stand-alone GPS receivers that feature the high performance U-blox 6 timing engine. The accuracy of the time pulse signal in the LEA-6T is approximately 30 ns, and the time-to-first-fix is 29 s. The clock module is an ultra-high stability MCXO from Vectron MX-503 with an accuracy of ±30 ppb in a temperature range of −20–75 ℃. The power consumption of the module is 12 mA at 3.3 V. For example, to generate the clock (f = 9,600 Hz) in the CSAMT mode, the MCU sets the division factor to 1280 and the CPLD divides the clock signal which is triggered by the PPS from the GPS module. After 50 PPS counts, the MCU sets the division factor to 1,600, and the output clock frequency changes to 7,680 Hz. The above steps are then repeated, and each division is triggered by the PPS.

The poorest condition for meeting the high phase precision requirement in the SIP mode is that the error is 30 ppb during the 15 min circulation, which will cause a time drift of 27 μs. This time drift results in a 21 mrad phase error at 128 Hz.

### 3.3 Chopper circuit

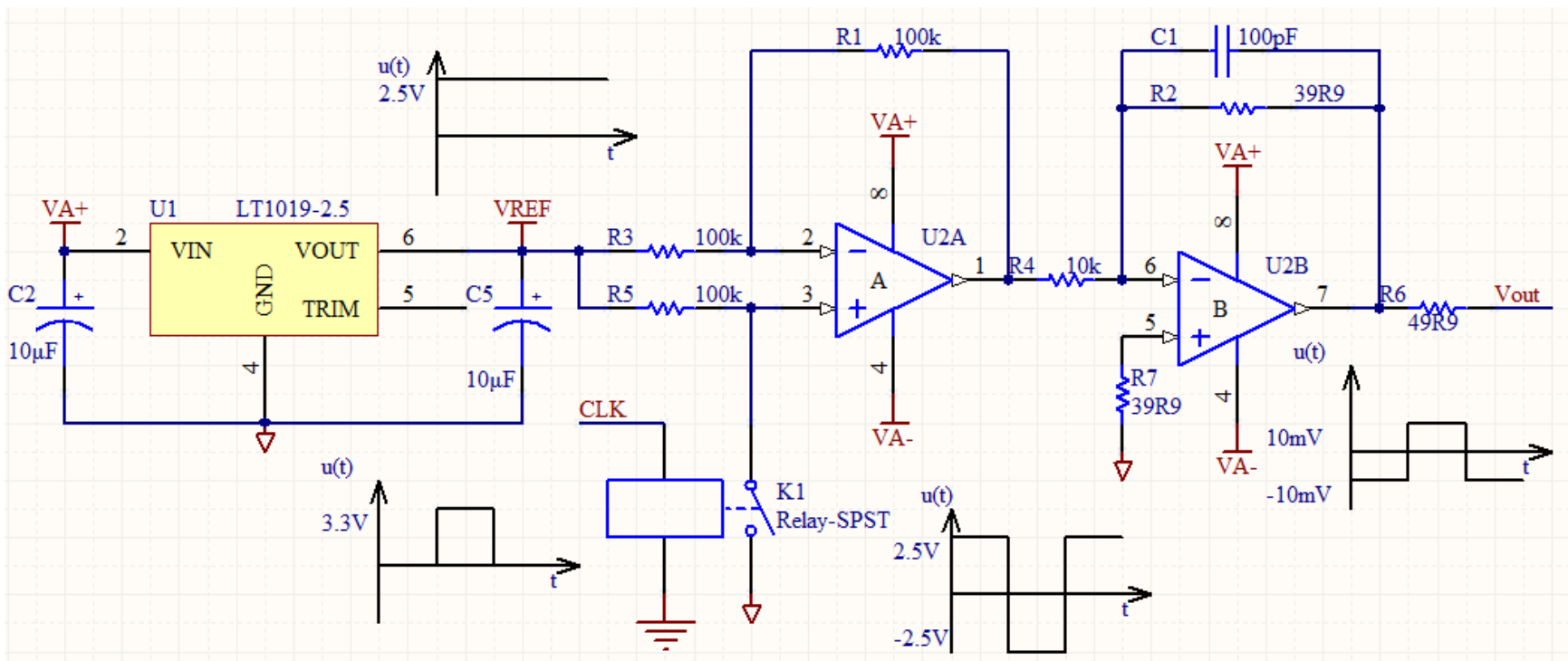

**Figure 7: Schematic of the chopper circuit (E channel).**

Figure 7 shows the schematic of the chopper circuit, which chops a high precision DC reference voltage into a bipolar square waveform driven by the switching clock. The circuit contains voltage references, a relay-SPST, and an amplifier. The voltage reference LT1019-2.5 (from Linear Technology) used is a 2.5 V high precision DC reference. The relay is driven by the switching clock, which is a 3.3 V CMOS square waveform. The switching clock control the connection U2 Pin3 and GND. The output from U2 Pin 1 is a bipolar square wave with an amplitude of 2.5 V. The component U2B is an attenuator and low pass filter, and the output decreases from 5 Vpp to 20 mVpp for the E channel. The bandwidth is limited to 100 kHz. The amplitude of the H channel is 200 mVpp, which is different from the gain of U2B.

### 3.4 Shaper circuit

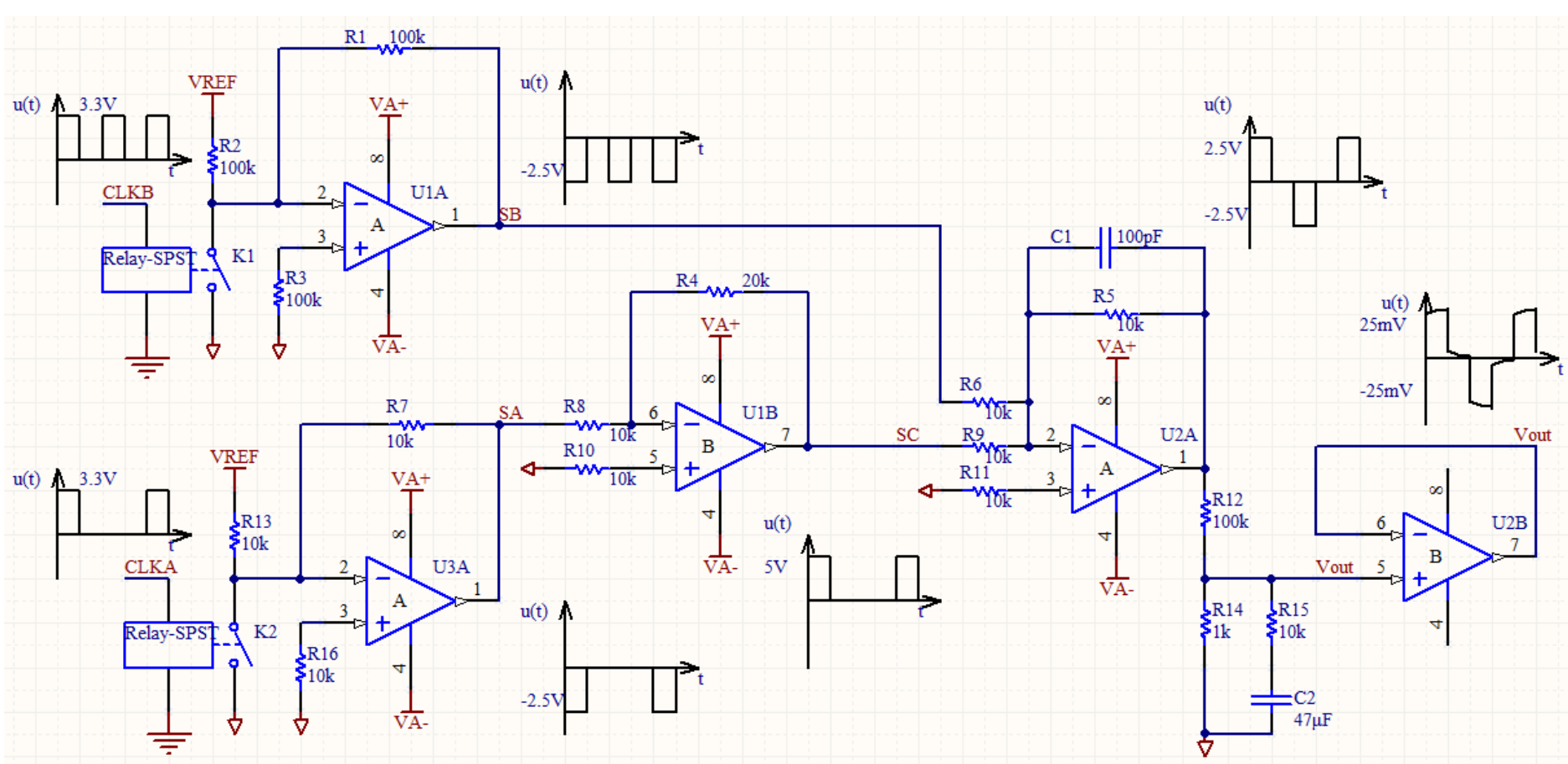

**Figure 8: Schematic of the shaper circuit.**

A schematic of the shaper circuit is shown in Fig. 8. The shaper circuit shapes the clock output from the CPLD into a PZNZ waveform. In the shaper circuit, there are two channel clock sources (CLKA and CLKB) and one output (Vout). The CLKA and CLKB outputs are set as different frequencies such that the frequency of CLKA is double the frequency of CLKB, and the duty ratio of CLKB is 1:3. The width of each pulse is 2 s. The amplifiers U1A and U3A are both used as those in the previous chopper circuit, which chops the DC reference to the square waveform. The clock is transformed into a square waveform with an amplitude of 2.5 V. The amplifier U1B is set with a gain of −2. The amplifier U2A adds nodes SB and SC. The waveform at node SD is a bipolar PZNZ waveform. The components R12, R14, R15, and C2 constitute an RC network to generate the PZNZ waveform based on the principles described earlier. The amplitude at the output of the shaper is approximately 50 mVpp.

## 4 Test

### 4.1 MT mode

We used the developed multifunction waveform generator to test our multifunction EM receiver (EMR6). The output of the multifunction waveform generator was connected to the input of the EMR6, which works in three sequential modes: 30 min for AMT mode, 24 h for MT mode, and 72 h for LMT mode. All raw data was processed using the white noise method by the MT data processing software (SSMT2000) from Phoenix Geophysics. The results of the data processing are shown in Fig. 9. The apparent resistivity and impedance phase are present across the entire nine-decade frequency range from 1 x $10^{-5}$ Hz to 1 x $10^{4}$ Hz. The theoretical simulation result of the apparent resistivity was 1,270 Ω m, and the impedance phase was approximately 0 °. When the experimental results were compared to those from the theoretical simulation, the bias error between the experiment results and theoretical value for the apparent resistivity was 1.5%.

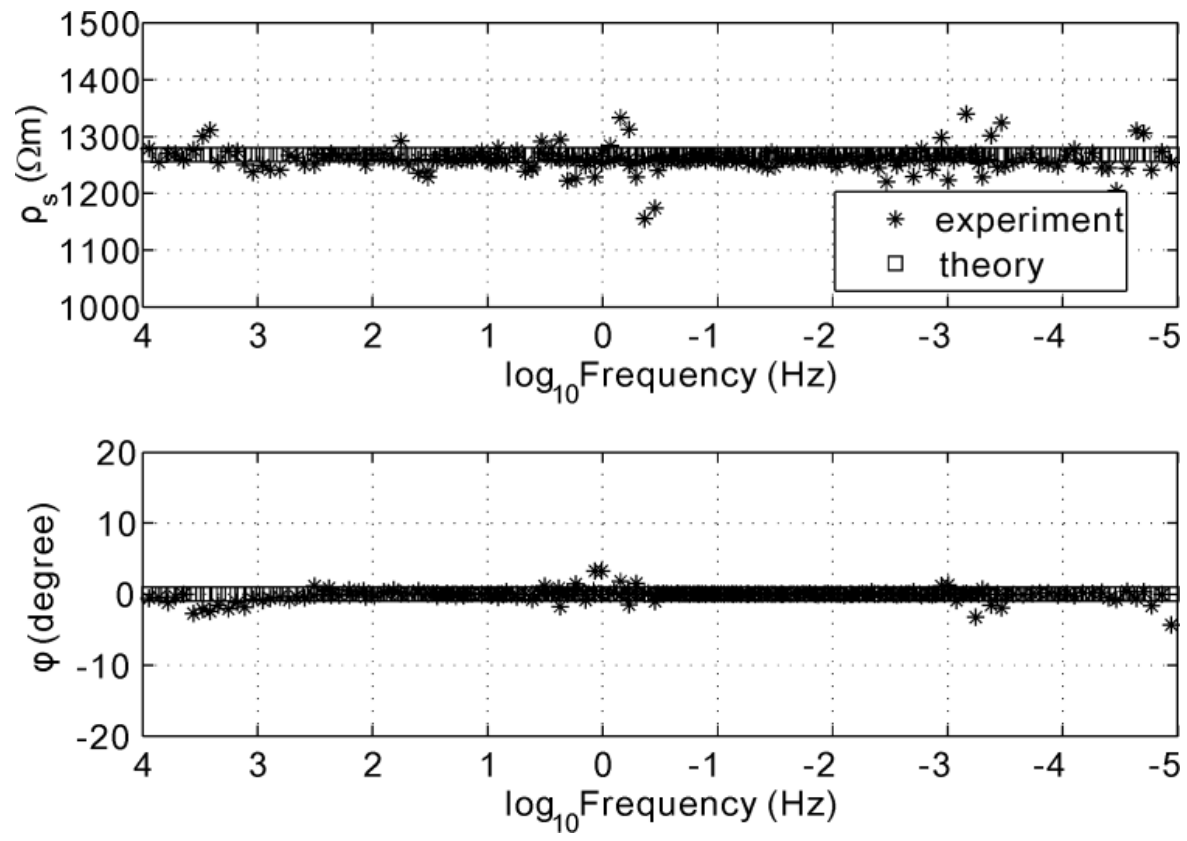

**Figure 9: Apparent resistivity and impedance phase results from the EMR6 testing. The upper subfigure shows the apparent resistivity curves, and the lower subfigure shows the impedance phase curves.**

The results of the experiment indicated that the multifunction waveform generator could test the MT receiver across the entire MT band, and the EMR6 receiver was shown to have a correct response in the target bands.

### 4.2 TDIP mode

To verify the TDIP waveform, the generator was switched to the TDIP mode and its output was connected to the EMR6 E channel input, and the EMR performs TDIP data acquisition during 2 min at 2,400 Hz sample rate. Figure 10 shows the time series captured by the EMR6. The full waveform was recorded for the entire time series of the E1 channel, including the primary and secondary field waveforms. The amplitude was approximately 50 mVpp, and the “on” time and “off” time pulse widths were 2 s.

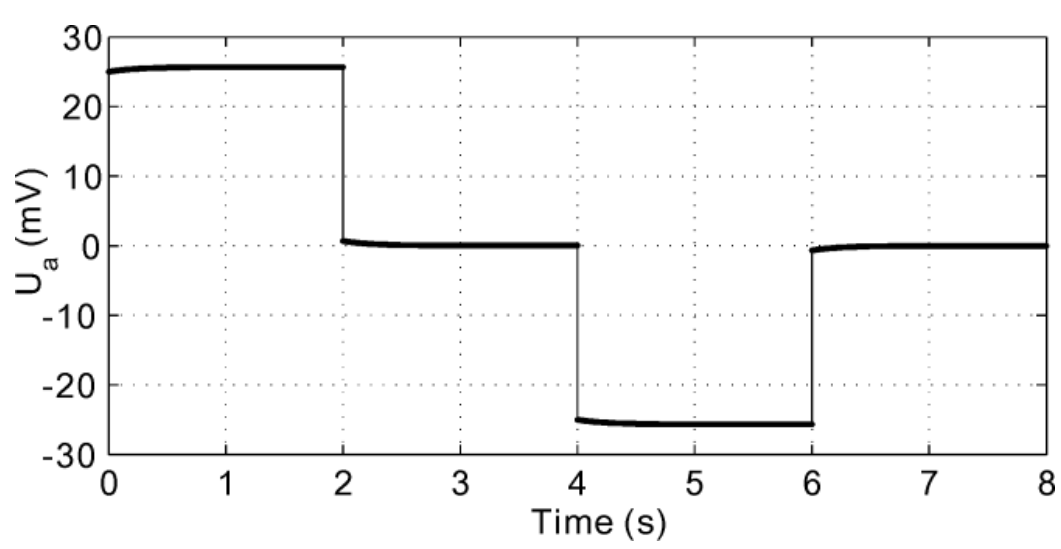

**Figure 10: Time series in TDIP mode as captured by the EMR6.**

The chargeability was calculated by dividing the secondary field into nine windows. The offset time of the first window was 10 ms, and each window had a width of 8, 16, 32, 64, 128, 256, 512, 1024, and

2048 sample points, respectively. Figure 11 shows a comparison between the measured results and those from the theoretical simulation. The chargeability was distributed from 0.7% to 9%. The results show that the output from the waveform generator during the experiment response was in good agreement with the theoretical output. The bias error of the nine windows was 0.8% between the measured results from the experiment and those from the theoretical simulation.

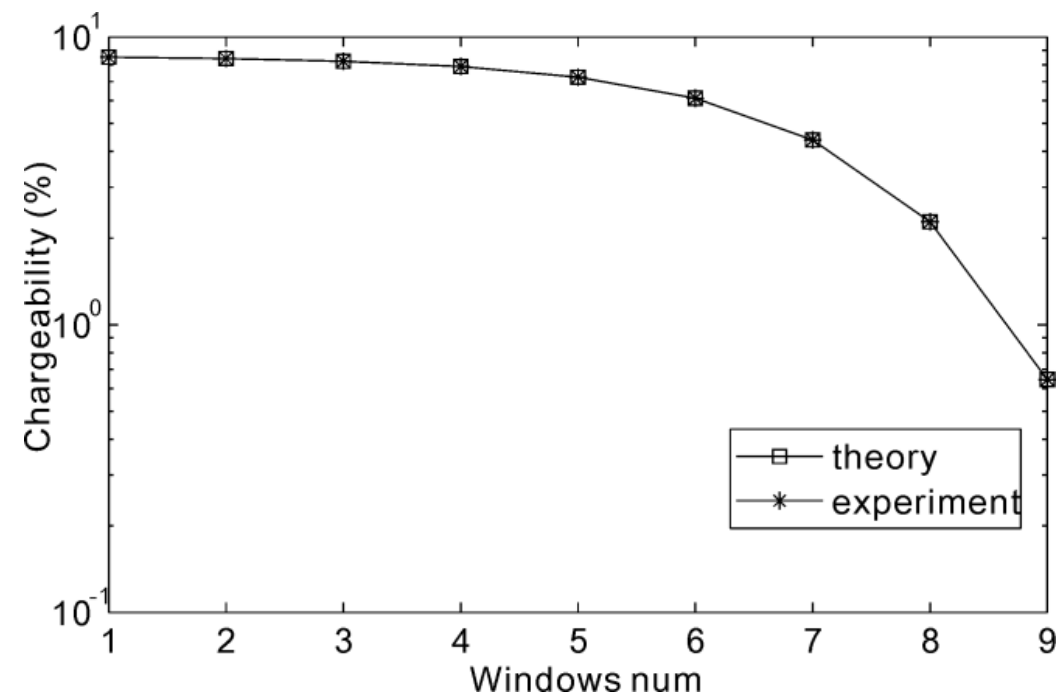

**Figure 11: Comparison between the calculated results and the theoretical simulation results for TDIP mode.**

### 4.3 SIP mode

The EMR6 and waveform generator operated on the SIP mode. To simplify the experiment, the current data of the transmitter and a variety of geometric factors were not considered. The EMR6 recorded the frequency-swept square waveform for 15 min and calculated the amplitude of each target frequency. Figure 12 shows the 12 target frequencies from 128 Hz to 0.0625 Hz. The amplitudes measured in the experiment were approximately 12.74 mV and the phases were approximately 0 °. The bias error between the experiment and theory was 0.5 % across the entire frequency range.

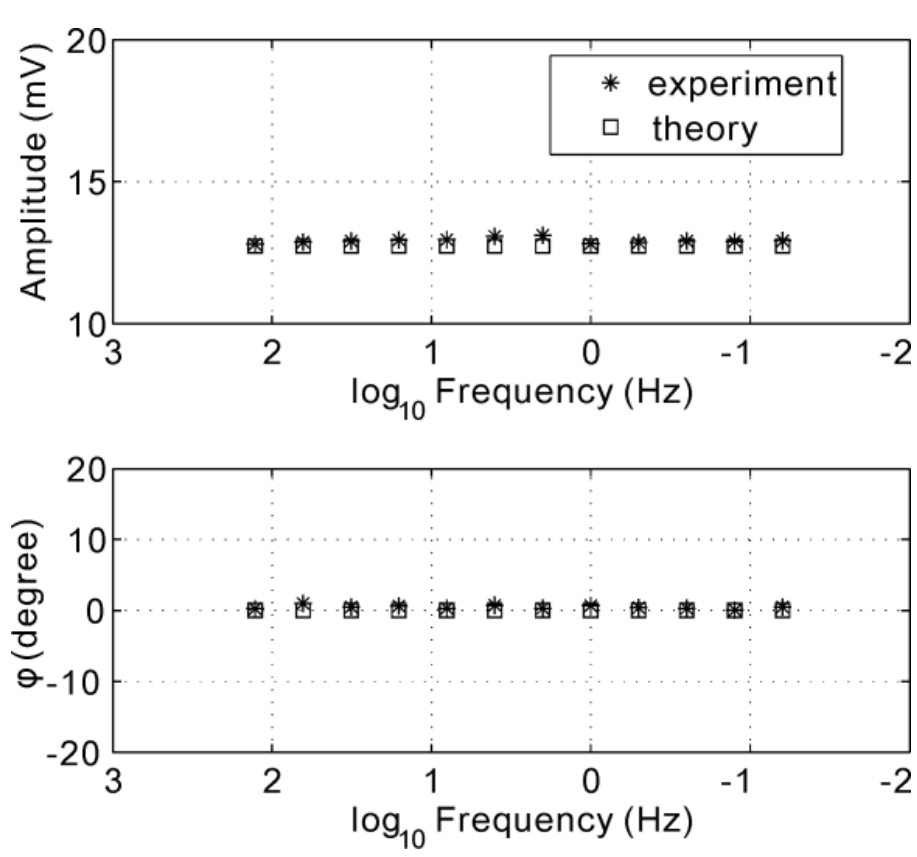

**Figure 12: Comparison between the experimental results and the theoretical simulation results for the SIP mode.**

### 4.4 CSAMT mode

The EMR6 and waveform generator operated on the CSAMT mode and recorded the E- and H-channel swept square waveforms. The Cagniard apparent resistivity and impedance phase were calculated. The receiver recorded the swept frequency waveform for 50 min. Figure 13 shows the results calculated for the 41 target frequency points from 9,600 Hz to 0.9375 Hz. The apparent resistivity measured in the experiment was approximately 1,268 Ω·m and the phase was approximately 0 °. The bias error between the experiment and theory was 1.3% across the entire frequency range.

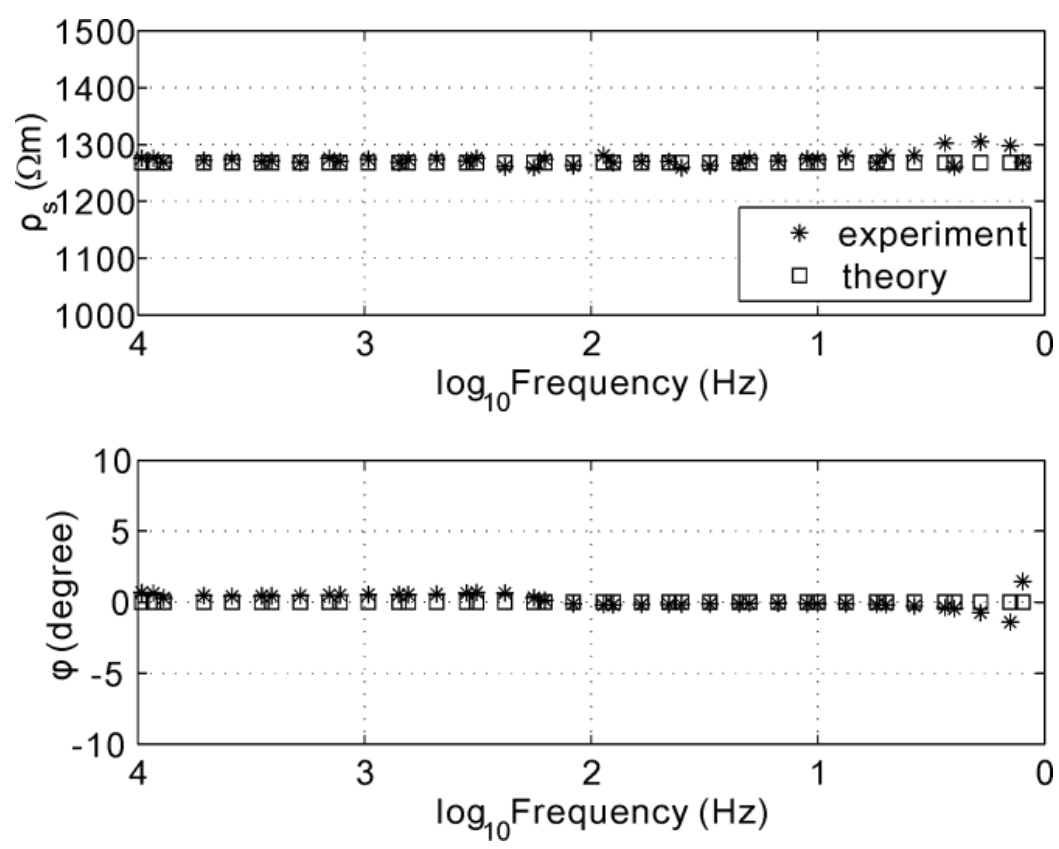

**Figure 13: Comparison between the calculated results and the simulation results for the CSAMT mode.**

### 4.5 Comparison with commercial function waveform generator

From the results of the above experiment, Table 2 presents the comparison of the specifications of the developed multifunction waveform generator and Agilent 33510B function waveform generator. The presented results indicate that Agilent 33510B is not suitable for EM receiver testing. The developed multifunction waveform generator is a better signal source for this purpose.

**Table 2. Specification comparison with Agilent 33510B.**

| Specifications | Agilent 33510B | Newly developed multifunction waveform generator |
|---|---|---|
| Function | Sine/square/ramp/pulse/triangle/PRBS/white noise | PRBS/Frequency sweep/PZNZ |
| Channels | 2 | 4 (Ex/Ey/Hx/Hy) |

| | | |
|---|---|---|
| Time sync | Internal timer or Ext Trig connector | GPS |
| PRBS | 1m bps~50 M bps | 10 μHz~100 kHz |
| Power | AC 100–240 V | Built-in rechargeable Li-ion battery |

## 5. Conclusion

The performance of the existing commercial function/arbitrary waveform generator is deficient with regard to time synchronization and waveform requirements. The multifunction waveform generator described in this paper was found to be useful for conducting EM receiver testing for multiple EM methods. The results of the testing show that the multifunction waveform generator could provide three mode signals containing independent broadband signals with different spectral characteristics, white noise, a repeating swept square waveform, and a PZNZ waveform consisting of primary and secondary fields. The apparent resistivity and impedance phase of the broadband white noise source was very flat across a wide frequency band. The theoretical design of the TDIP waveform was shown to have a correct response. In the SIP and CSAMT modes, the results of the experiment and theoretical simulation were a close match. In addition, the generator had other advantages in that it was easy to use and had low power consumption.

Furthermore, various parameters, such as the programmable frequency step schedule in the CSAMT and SIP modes and the pulse width of the TDIP waveform, are configurable by the user through the hardware interface or by loading the memory.

## Author Contributions

Kai Chen developed the required hardware and software. Sheng Jin created the overall design and performed the tests. Ming Deng was the chopper and shaper circuit technology consultant.

## Competing Interests

The authors declare that they have no conflict of interest.

## Acknowledgments

General funding was provided by the National High Technology Research and Development Program of China (2014AA06A603), National Science Foundation of China (61531001), Central University

Fundamental Research Project of the Ministry of Education (2652015403), and Key Development Program of China (2016YFC0303100). We are thankful for the data processing software from Phoenix Geophysics. We would also like to thank Editage [www.editage.cn] for their English language editing services.

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
