# Peer review of "Multifunction waveform generator for EM receiver testing"

_Geoscientific Instrumentation, Methods and Data Systems, 2017_

## Referee Comment (RC1) · Anonymous Referee #1 · 14 Nov 2017

The paper describes a step forward in advancing EM generator technology for testing EM receivers. The content is appropriate for publication. The paper is generally well written and thought out although there are a small number of mistakes. The structure of the paper needs some attention. Details of the paper's structural issues and a possible solution are included below.

1. The introduction seems to be too long. For example lines 56 to 64 seems to describes the solution to the problem too soon and would be better placed in a discussions section.

2. Section 4.5 does not really belong in the results section as it is comparing specifications of EM generators and is not in itself a result.

[Figure]

I recommend a small amount of rearranging of the paper to make it easier to read. I would suggest creating a new section 6 for the conclusions. Then section 5 could be a discussions section which combines lines 56 to 64, from the introduction, together with the text and table from section 4.5.

Some other issues:

In the introduction it is not easy to identify the motivation for the work, e.g. the requirements are a bit lost within some discussions on previous testing of the EMR6. Starting a new paragraph for the sentence at line 50 in the paper would help.

At the end of the introduction there needs to be a paragraph with a brief description of the content of each section.

Line 42 Should it be "... on the surface and in tunnels"? Lines 204-206 Needs some editing because the text is a bit difficult to follow Line 214 sentence doesn't make sense Line 226 delete 'those'

---

## Short Comment (SC1) · 15 Nov 2017

Dear Anonymous Referee, Thank you for your comment and for valuable remarks on the manuscript! According to your advice, we revised this manuscript. All of the changes have been made using red text as the supplement.

1) We have creating a new section 5 for the conclusions which combines lines 56 to 64, from the introduction, together with the text and table from section 4.5. 2) We have started a new paragraph for the sentence at line 50. 3) At the end of the introduction, a brief description of the content of each section is added. "In this paper, we describe a multifunction waveform generator. The details of the waveform generator, including both the system design and the hardware principles are discussed. Section II begins

by describing the requirement of the three types of waveforms. Next, we discuss the hardware principle design in Section III. Section IV follows, describing the result of test to confirm the function of the waveform generator. In Section V, discussion and specification comparison table are present." 4) Line 42 has corrected as "... on the surface and in tunnels" from "on the surface and tunnels". 5) Line 204-206 are revised as: In the SIP mode, an accuracy of $\pm 30$ ppb clock signal will drift 27 $\mu$s, while the whole circulation last 15min. And the time drift error will cause 21 mrad phase error while the frequency of transmitter is set as 128 Hz. For high phase precision, the frequency error of the switching clock signal must be as low as possible. 6) Line 214 add "The switching clock is divided from the above MCXO." 7) "those in the previous" form Line 226 has been deleted.

Once again, thank you very much for your comments and suggestions. We tried our best to improve the manuscript and we made all necessary changes in the manuscript. We truly appreciate the time and efforts of the editors and reviewers, and we sincerely hope that our corrections will meet your approval.

Please also note the supplement to this comment:
https://www.geosci-instrum-method-data-syst-discuss.net/gi-2017-43/gi-2017-43-SC1-supplement.pdf

**Supplement:**

[revised manuscript text omitted]

---

## Referee Comment (RC2) · R. Chen (Referee) · 27 Nov 2017

The paper introduces the design and the testing of a customized multi-function wave-form generator for the testing of an EM receiver. There are very few literatures considering this problem although many EM instrument owns build-in waveform generator for testing. I suggest the publishing of the manuscript if the following suggestions are considered by authors .

1. This manuscript lacks essential references considering the testings of EM receiver. There are papers and patents in English and Chinese considering this problems. This problems is attacked by many Chinese authors. I just list a few of works by myself here.

Ruijie Shen, Kailin Qiu, Hongchun Yao, Zhanxiang He, Xianghong Lu, Rujun Chen:

[Figure]

The design and realization of the signal generator for the calibration of MT instrument based on FPGA and GPS synchronizationïïjĹin Chinese with English abstract). DOI:10.11720/wtyht.2013.1.15

Hongchun Yao, Rujun Chen, Shengli Liu: Development of automatic test software for multi-frequency IP receiver based on the Ethernet (in Chinese with English Abstract). Wutan Huatan Jisuan Jishu 03/2010; 32:211-216.

Rujun Chen, He Zhangxiang, Qiu Jieting, He Lanfang, Cai Zixing: Distributed data acquisition unit based on GPS and ZigBee for electromagnetic exploration. Conference Record - IEEE Instrumentation and Measurement Technology Conference 01/2010;, DOI:10.1109/IMTC.2010.5488223

He Zhanxiang, Chen Rujun, Liu Xuejun, He Lanfang: Magnetotelluric instrument performance evaluation method. Ref. No: CN200910237810, Year: 05/2011

Chen Rujun, Yao Hongchun, Liu Shengli: Automatic testing system of pseudo-random multi-frequency instrument receiver. Ref. No: CN200910044457, Year: 03/2010

Chen Rujun, Luo Weibing, He Jishan, Yan Liang, Liu Shi: Multi-functional transmitter and controller for electrical surveying signal. Ref. No: CN200610032163, Year: 02/2007

2. The authors should summary the works by other peoples in the subject studied by the authors.

3. In line 10 - line 11, the time synchronization of waveform generator is not needed for the testing of EM receiver (AMT, MT, CSAMT, SIP, etc.). The time sync is only needed for transmitter, receiver, and calibration. We measure E/H or voltage/current at the same time. This process doesn't need precise sync of input signal. Commercial waveform generator is OK for the testing of EM receiver.

4. In line 41, simple introduction of EMR6 is needed.

5. In line 151, specification of RTC is needed.

6. Section 3.1, exact types of MCU and CPLD are needed.

7. Line 171, What's the meaning of COMS?

8. Line 173, AMY may be AMT.

9. Line 181-185, it's not clear about the process of testing using GPS time. 10. Line 201-203, it seems no stop time in frequency switching. MCU needs time to control CPLD and CPLD needs time to sync and response. What's the communication method between CPLD and MCU? 11. THD is a vital factor determining the performance of EM receiver. Just like seismic instrument, THD testing is vital important. The authors don't consider this problem in the testing of EM receiver.

Rujun Chen, Associate Professor

1.School of Geosciences and Info-Physics, Central South University, 410083, Changsha, China

2.Key Laboratory of Metallogenic Prediction of Nonferrous Metals and Geological Environment Monitoring, Ministry of Education (Central South University), 410083,Changsha, China

3.Key Laboratory of Non-Ferrous Resources and Geological Hazard Detection, 410083, Changsha, China

---

## Author Comment (AC2) · 4 Dec 2017

Dear Professor Chen, Thank you for your comment and for your valuable remarks on the manuscript! We have revised this manuscript according to your advice. All changes have been made using red-colored font.

1) Comment from the Reviewer: This manuscript lacks essential references considering the testing of EM receiver. There are papers and patents in English and Chinese considering this problems. This problems is attacked by many Chinese authors. I just list a few of works by myself here. Author's response: Thank you for pointing this out. He et al. and Chen et al. have proposed some methods for MT instrument performance evaluation. We have added the references to these two studies

in the main text and in the references section. Author's changes in the manuscript: The following studies have been added as references. Chen, R.J., Yao, H.C., and Liu, S.L.: Automatic testing system of pseudo-random multi-frequency instrument receiver. Ref. No: CN200910044457, Year: 03/2010. He, Z.X., Chen, R.J., Liu, X.J., L., and He, L.F.: Magnetotelluric instrument performance evaluation method. Ref. No: CN200910237810, Year: 05/2011.

2) Comment from the Reviewer: The authors should summary the works by other peoples in the subject studied by the authors. Author's response: We agree that this manuscript lacks essential references related to the testing of EM receivers. Author's changes in the manuscript: The following sentence has been added in the second paragraph of the introduction. For the realization of a multi-frequency EM receiver, (Chen et al., 2009) developed an automatic testing system based on pseudo-random signals. For MT receiver performance evaluation, (He et al., 2011) conducted a series process method with high efficiency. Which as detailed in the patents applied on several aspects of instrument testing."

3) Comment from the Reviewer: In line 10 - line 11, the time synchronization of waveform generator is not needed for the testing of EM receiver (AMT, MT, CSAMT, SIP, etc.). The time sync is only needed for transmitter, receiver, and calibration. We measure E/H or voltage/current at the same time. This process doesn't need precise sync of input signal. Commercial waveform generator is OK for the testing of EM receiver. Author's response: I apologize for not clarifying this in the original manuscript. For the MT/AMT mode, this process does not need the precise synchronization of the input signal. Nevertheless, I should have explained that the time synchronization of the waveform generator is important in the CSAMT and SIP modes. The waveform generator is a quasi-transmitter for indoor testing. The receiver measures the E/H or voltage/current signal that the GPS time information uses as the time reference. If there is no time reference, the schedule list of the waveform generator cannot function at all. Moreover, the commercial waveform generator is insufficient with regard to time synchronization.

Line 11 – Line 13 present the drawback of the current commercial waveform generator for EM receiver testing. Author's changes in the manuscript: Line 10 has been added in the manuscript as follows. "In controlled source audio magnetotelluric (CSAMT) and spectrum induced polarization (SIP) mode testing, the waveform generator should use the GPS time as a reference for repeating schedule."

4) Comment from the Reviewers: In line 41, simple introduction of EMR6 is needed. Author's response: A simple introduction of EMR6 has been added. Author's changes in the manuscript: The first sentence of the second paragraph in the introduction was changed to "EMR6 is a new multifunctional EM receiver for deep metal mineral exploration EM survey, which was developed by China University of Geosciences (Beijing) (CUGB) and which supports the audio magnetotelluric (AMT), MT, CSAMT, SIP, and TDIP methods on the surface and in tunnels."

5) Comment from the Reviewer: In line 151, specification of RTC is needed. Author's response: The specification of the RTC was added. Author's changes in the manuscript: The following sentences were added to the first paragraph of Section 3.1. "DS3231 from Maxim Integrated is a low-cost, extremely accurate RTC with an integrated temperature compensated crystal oscillator (TCXO) and crystal. It has the advantages of high integration level, low power, and ease of use."

6) Comment from the Reviewer: Section 3.1, exact types of MCU and CPLD are needed. Author's response: The exact types of the MCU and CPLD have been added. Author's changes in the manuscript: The following sentences were added to the first paragraph of Section 3.1. "The low power 8-bit micro-controller MSP430G2553 from Texas Instruments was used as the MCU and 5M80ZE64 from Altera was used as the CPLD."

7) Comment from the Reviewer: Line 171, what's the meaning of COMS? Author's response: It is a typo; it should be "CMOS." We have corrected the typo. Author's changes in the manuscript: "COMS" was changed to "CMOS."

8) Comment from the Reviewer: Line 173, AMY may be AMT. Author's response: It is a typo; it should be "AMT." We have corrected the typo. Author's changes in the manuscript: "AMY" was changed to "AMT."

9) Comment from the Reviewer: Line 181-185, it's not clear about the process of testing using GPS time. Author's response: For locking the GPS time to the local RTC, the MCU obtains the time information from the GPS module and writes the time information to the RTC. In the CSAMT, SIP, and TDIP modes, the MCU must know the current time before it can switch the repeating schedule. We have added the following explanation of the GPS time at the end of Section 3.1. Author's changes in manuscript: At the end of Section 3.1, we have added, "For locking the GPS time to the local RTC, the MCU obtains the time information from the GPS module and writes it to the RTC. In the CSAMT, SIP, and TDIP modes, the MCU must know the current time before it can switch the repeating schedule."

10) Comment from the Reviewer: Line 201-203, it seems no stop time in frequency switching. MCU needs time to control CPLD and CPLD needs time to sync and response. What's the communication method between CPLD and MCU? Author's response: I apologize for not clearly presenting this part in the original manuscript. While switching to a different divider, there is a one-second pause before the start of the next divider. The communication method between the CPLD and MCU is a serial-peripheral interface. Author's changes in the manuscript: "There is a one-second pause between each switch divider for the CPLD to synchronize with the MCU. The CPLD is controlled by an MCU with a serial-peripheral interface." These sentences were added to the first paragraph of section 3.2.

11) Comment from the Reviewer: THD is a vital factor determining the performance of EM receiver. Just like seismic instrument, THD testing is vital important. The authors don't consider this problem in the testing of EM receiver. Author's response: The THD is very important in the performance of the EM receiver. Actually, in the indoor test, we used a low distortion signal generator (DS360) from Stanford Research Systems as a

pure signal source. The realization of this function is challenging. It is not contained in our current work, and may be realized in a future development. Author's changes in the manuscript: At the end of the discussion section, we have added, "Total harmonic distortion (THD) is another important factor determining the performance of the EM receiver. We have used a low distortion signal generator (DS360) from Stanford Research Systems as a pure signal source. This function is not contained in the current work, and it may be realized in a future development."

We look forward to hearing from you regarding our submission. We would be glad to respond to any further questions and comments that you might have.

Sincerely, Kai Chen China University of Geosciences Beijing 100083 ck@cugb.edu.cn

Please also note the supplement to this comment:
https://www.geosci-instrum-method-data-syst-discuss.net/gi-2017-43/gi-2017-43-AC2-supplement.pdf

**Supplement:**

**Multifunction waveform generator for EM receiver testing**

Kai Chen, Sheng Jin, Ming Deng

China University of Geosciences, Beijing, China

*Correspondence to:* Kai Chen (ck@cugb.edu.cn)

[revised manuscript text omitted]

$$u_1(t) = \begin{cases} A, & 0 \le t < T/4 \\ 0, & T/4 \le t < T/2 \\ -A, & T/2 \le t < 3T/4 \\ 0, & 3T/4 \le t < T \end{cases} \qquad (2)$$

$$u_2(t) = \begin{cases} B - Be^{-t/\tau}, & 0 \le t < T/4 \\ Be^{-(t-\frac{T}{4})/\tau}, & T/4 \le t < T/2 \\ Be^{-(t-\frac{T}{2})/\tau} - B, & T/2 \le t < 3T/4 \\ -Be^{-(t-\frac{3T}{4})/\tau}, & 3T/4 \le t < T \end{cases}, \qquad (3)$$

where $u_1(t)$ and $u_2(t)$ denote the primary and secondary electrical fields, respectively, and the exponent attenuation curve refers to the secondary electrical field.

[Figure]

**Figure 5: Schematic of the PZNZ waveform generator.**

Figure 5 shows a schematic of the circuit used to generate the PZNZ waveform, which contains both primary and secondary electrical fields. The resistor-capacitor (RC) network transforms the PZNZ ($V_{in}$)

waveform into a PZNZ waveform ($V_{out}$) with the secondary electrical field.

$$A = \frac{R_3//R_2}{R_1 + R_3//R_2} K \qquad (4)$$

$$B = \frac{R_3//R_2}{R_1 + R_3//R_2} K - \frac{R_2}{R_1 + R_2} K \qquad (5)$$

$$\tau = R_3 C_1 \qquad (6)$$

The amplitude of Vin is ±K. A and B used in Eqs. (2) and (3) can be calculated from Eqs. (4) and (5).

The time constant (τ) is given by Eq. (6).

**3 Hardware principle**

**3.1 Block diagram**

Figure 6 presents a block diagram of the waveform generator, which has the advantages of automation, ease of use, high phase precision, and low power. As shown in the figure, the hardware consists of an input switch, microcontroller unit (MCU), GPS module, complex programmable logic device (CPLD), compensated microprocessor crystal (MCXO), real time clock (RTC), chopper, shaper, multiplexer, power conversion circuit, and built-in Li-ion battery package. DS3231 from Maxim Integrated is a low-cost, extremely accurate RTC with an integrated temperature compensated crystal oscillator (TCXO) and crystal. It has the advantages of high integration level, low power, and ease of use. The low-power 8-bit micro-controller MSP430G2553 from Texas Instruments was used as the MCU and

5M80ZE64 from Altera was used as the CPLD. The input switch is used to change the work mode without configuring the external complicated parameters. The LED is used to indicate the working status. The different work modes output different LED flash patterns. The GPS module is LEA-6T

from U-blox, which provides a high precision time pulse per second with low power consumption. The

MCXO has the specifications of high stability clock source (12.288 MHz) (±30 ppb) and low power consumption (3.3 V &12 mA). To lock the GPS, the MCU receives the time information from the GPS

module and writes to the RTC. The CPLD is used to implement a frequency divider, logic operator,

PPS lock, and tracking. The RTC is the time counter used for circulation of the frequency stepping schedule. The chopper circuit chops a high precision DC reference into a bipolar square waveform under the control of the CPLD. The shaper generates the TDIP PZNZ waveform. Moreover, the sum of the primary and secondary field signals is also provided as output. The multiplexer is controlled by the

MCU to select either the chopper or the shaper output. The power module converts the Li-ion battery (11.1 V and 10 Ahr) voltage to digital power at 3.3 V and analogue power at ±3 V.

[Figure]

**Figure 6: Block diagram of the multifunction waveform generator circuit.**

When the MT mode is selected, the MCU controls the CPLD to generate two independent PRBS. The outputs Ex and Hy share one PRBS, whereas Ey and Hx share another PRBS. The two PRBSs are different for different phases. The chopper circuit converts the 3.3-V CMOS PRBS into a bipolar

±10-mV square wave for Ex and Ey. The amplitude of Hx and Hy is ±100 mV. The MCU uses the multiplexer output to select the chopper output. Based on the different modes (AMT/MT/LMT), the output selected by the input switch and the smallest code width of the PRBS is changed. To decrease the power consumption, the GPS module is powered down during the MT mode.

When the CSAMT mode is selected, the MCU reads the GPS time information and writes to the

RTC. The MCU controls the CPLD divisionde factor according to the frequency stepping schedule.

The CPLD divides the clock source (12.288 MHz) to the target frequency step by step. The divider is trigged by the PPS from the GPS module. The 41 frequency point clock signal drives the chopper circuit to generate a bipolar square waveform. All frequency stepping schedules start from the reference time base of 00:00:00 and repeat from high frequency to low frequency. For example, after the power is turned on and the GPS is locked, the current time is 02:20:00, the period of frequency stepping schedule is 50 min, the residual time is 40 min, the first output signal is the No. 40 frequency point (1.25 Hz), and the output is the rest 10 min of the current schedule. The frequency stepping schedule is repeated continually.

The procedure in the SIP mode is the same as that described above for the CSAMT module for the step schedule with time synchronization. A typical frequency stepping schedule period is 15 min, and contains 12 frequency points.

When the TDIP mode is selected, the RTC time is locked to the GPS module and the CPLD generates pulse with 2-s width synchronization as the PPS from the GPS module to drive the shaper circuit. The reference time base is 00:00:00, and the typical period is 8 s with duty ratio 1:1.

For locking the GPS time to the local RTC, the MCU obtains the time information from the GPS

module and writes it to the RTC. In the CSAMT, SIP, and TDIP modes, the MCU must know the current time before it can switch the repeating schedule.

**3.2 Clock source**

In accordance with the high precision phase requirement from the CSAMT and SIP modes, an integrated U-blox GPS module and MCXO were selected as the high stability clock source. The

LEA-6T module series is a family of stand-alone GPS receivers that feature the high performance

U-blox 6 timing engine. The accuracy of the time pulse signal in the LEA-6T is approximately 30 ns, and the time-to-first-fix is 29 s. The clock module is an ultra-high stability MCXO from Vectron

MX-503 with an accuracy of ±30 ppb in a temperature range of −20–75 ℃. The power consumption of the module is 12 mA at 3.3 V. For example, to generate the clock (f = 9,600 Hz) in the CSAMT mode, the MCU sets the division factor to 1280 and the CPLD divides the clock signal, which is triggered by the PPS from the GPS module. After 50 PPS counts, the MCU sets the division factor to 1,600, and the output clock frequency changes to 7,680 Hz. The above steps are then repeated, and each division is triggered by the PPS. There is a one-second pause between each switch divider for the CPLD to synchronize with the MCU. The CPLD is controlled by an MCU with a serial-peripheral interface.

In the SIP mode, a clock signal with an accuracy of ±30 ppb will drift by 27 μs, while the entire circulation lasts 15 min. The time drift error will cause a 21-mrad phase error when the frequency of the transmitter is set to 128 Hz. For high phase precision, the frequency error of the switching clock signal must be as low as possible.

**3.3 Chopper circuit**

[revised manuscript text omitted]

Total harmonic distortion (THD) is another important factor determining the performance of the EM

receiver. We have used a low distortion signal generator (DS360) from Stanford Research Systems as a pure signal source. This function is not contained in the current work, and it may be realized in a future development.

**6. Conclusion**

The performance of the existing commercial function/arbitrary waveform generator is deficient with regard to time synchronization and waveform requirements. The multifunction waveform generator described in this paper was found to be useful for conducting EM receiver testing for multiple EM

methods. The results of the testing show that the multifunction waveform generator could provide three mode signals containing independent broadband signals with different spectral characteristics, white noise, a repeating swept square waveform, and a PZNZ waveform consisting of primary and secondary fields. The apparent resistivity and impedance phase of the broadband white noise source was very flat across a wide frequency band. The theoretical design of the TDIP waveform was shown to have a correct response. In the SIP and CSAMT modes, the results of the experiment and theoretical simulation were a close match. In addition, the generator had other advantages in that it was easy to use and had low power consumption.

Furthermore, various parameters, such as the programmable frequency step schedule in the CSAMT

and SIP modes and the pulse width of the TDIP waveform, are configurable by the user through the hardware interface or by loading the memory.

**Author Contributions**

Kai Chen developed the required hardware and software. Sheng Jin created the overall design and performed the tests. Ming Deng was the chopper and shaper circuit technology consultant.

**Competing Interests**

The authors declare that they have no conflict of interest.

**Acknowledgments**

General funding was provided by the National High Technology Research and Development Program of China (2014AA06A603), National Science Foundation of China (61531001), Central University Fundamental Research Project of the Ministry of Education (2652015403), and Key Development Program of China (2016YFC0303100). We are thankful for the data processing software from Phoenix Geophysics. We would also like to thank Editage [www.editage.cn] for their English language editing services.